# Therapeutic Cancer Vaccination with Ex Vivo RNA-Transfected Dendritic Cells—An Update

**DOI:** 10.3390/pharmaceutics12020092

**Published:** 2020-01-23

**Authors:** Jan Dörrie, Niels Schaft, Gerold Schuler, Beatrice Schuler-Thurner

**Affiliations:** Department of Dermatology, Universitätsklinikum Erlangen, Friedrich-Alexander-Universität Erlangen-Nürnberg, Hartmannstraße 14, 91052 Erlangen, Germany; jan.doerrie@uk-erlangen.de (J.D.); niels.schaft@uk-erlangen.de (N.S.); gerold.schuler@uk-erlangen.de (G.S.)

**Keywords:** therapeutic vaccination, dendritic cells, RNA, electroporation, clinical trial, immunotherapy

## Abstract

Over the last two decades, dendritic cell (DC) vaccination has been studied extensively as active immunotherapy in cancer treatment and has been proven safe in all clinical trials both with respect to short and long-term side effects. For antigen-loading of dendritic cells (DCs) one method is to introduce mRNA coding for the desired antigens. To target the whole antigenic repertoire of a tumor, even the total tumor mRNA of a macrodissected biopsy sample can be used. To date, reports have been published on a total of 781 patients suffering from different tumor entities and HIV-infection, who have been treated with DCs loaded with mRNA. The majority of those were melanoma patients, followed by HIV-infected patients, but leukemias, brain tumors, prostate cancer, renal cell carcinomas, pancreatic cancers and several others have also been treated. Next to antigen-loading, mRNA-electroporation allows a purposeful manipulation of the DCs’ phenotype and function to enhance their immunogenicity. In this review, we intend to give a comprehensive summary of what has been published regarding clinical testing of ex vivo generated mRNA-transfected DCs, with respect to safety and risk/benefit evaluations, choice of tumor antigens and RNA-source, and the design of better DCs for vaccination by transfection of mRNA-encoded functional proteins.

## 1. Tumors and the Immune System

Therapeutic cancer vaccination is a concept for treating tumor patients by immunizing them against their own tumor. As early as 1891, the application of bacterial substances (which we now know to have served as adjuvants) into tumors was executed by William Coley, who achieved a clinical response rate of 10% in soft tissue sarcoma [1,2]. During the 20th century, chemotherapy and radiation therapy were developed and superseded immunotherapy. The concept of immune surveillance, however, was later resumed and pursued [3,4] (and reviewed by [5,6]). The striking success that was achieved in preventive vaccination against infectious diseases suggested that the immune system could be utilized against malignancies in a similar fashion.

## 2. Therapeutic and Preventive Vaccination

There are, however, crucial differences between a preventive vaccine against an infectious disease and a therapeutic vaccination against an existing cancer. Firstly, the malignant cells arise from the body’s own healthy cells—therefore, the immune system’s primary function to distinguish self from foreign is circumvented. Secondly, the malignancy has developed in the presence of a functioning immune system and has hence adapted to immune surveillance. Thirdly, most tumors influence the immune system in their favor. Due to these circumstances, the induction of an effective anti-tumor immunity resembles that of a selective autoimmunity under arduous conditions. This necessitates the use of a highly immunogenic vaccination strategy, able to break tolerance and overcome immune suppression.

## 3. Dendritic Cells as Cancer Vaccine

The mammalian cell type which is specialized in initiating all adaptive immune responses was described in 1973 by Ralph Steinman and Zanvil Cohn and termed dendritic cells (DCs) [7]. They are considered to be sentinels of the immune system [8,9], and watch over the induction of immune responses and the maintenance of tolerance. They are also considered to be the link between innate and adaptive immunity [10].

The ability of DCs to induce tumor regression in murine models was shown more than two decades ago [11,12]. The first application in human beings was reported in 1996 [13], although the cells used in this study did not constitute highly enriched real DCs according to standards set later on [14], and a first practical protocol for generating human DCs in sufficient numbers from blood monocytes was also published in 1996 [15]. This opened up the possibility of a broader clinical application of DCs, and other encouraging reports soon followed [16,17]. Immune responses induced by the vaccination were later shown to correlate with clinical results, notably if monitored in tissues [18]. Although alternative protocols to generate DCs followed [19,20,21], the majority of clinical trials used this original protocol to generate monocyte-derived DCs (moDCs) by incubation over 5 to 8 days with GM-CSF and IL-4 [10,15] (see Table 1).

### 3.1. Dendritic Cell Maturation

To become immunogenic, a DC needs to undergo a process termed maturation [22]. This complex program is naturally induced by an exposure of the DC to danger signals [23]. It involves various phenotypic changes, including the up-regulation of co-stimulatory surface markers and the secretion of pro-inflammatory cytokines. The DCs’ ability to migrate toward peripheral lymph nodes, where they encounter the naïve T cells, also depends on the DCs’ maturation state [24]. It is, therefore, not surprising that maturation is an absolute prerequisite for the immunogenicity of DCs in vivo in humans [24]. Immature DCs were shown to even induce tolerance [25].

In moDCs, the maturation can be triggered with monocyte-conditioned medium [26]. This can be replaced with a cocktail of cytokines consisting of IL-1ß, IL-6, TNF, and PGE_2_ [27], termed MCM-mimic (MCMM) [26]. As shown in Table 1, the MCMM cocktail was used in the majority of published clinical trials, but other cocktails, including, e.g., IFNγ or TLR agonists, but also TNFα alone or combined with PGE_2_ only, have been used.

### 3.2. Loading Dendritic Cells with Antigen

The most straightforward way to load DCs with tumor antigen-derived epitopes is the direct pulsing with synthetic peptides, which was shown already in the initial DC trials to induce the corresponding immune responses [17]. This, however, requires the patient to be of an HLA haplotype for which suitable epitopes exist. An alternative is the use of larger parts of the antigen, i.e., long peptides [28] or even full-length proteins, which are then processed and cleaved by the DC’s endogenous antigen processing machinery. This results in the generation and presentation of relevant natural epitopes, which are contained in the antigen and fit into the HLA molecules of the DC, and thus also the patient’s HLA repertoire.

### 3.3. RNA-Transfection of Dendritic Cells

The easiest way to achieve MHC class I-restricted presentation in this fashion is the intracellular expression of the antigens, and a method considered ideal for clinical application is mRNA transfection [76]. Several forms of RNA transfection have been used by different groups over the past decades, e.g.: passive pulsing of mRNA, i.e., simple co-incubation of mRNA with the DCs, lipid-mediated transfection of mRNA, and mRNA electroporation, i.e., transfer of mRNA molecules through the cell membrane by an electrical pulse (reviewed in [77]), with the latter being used by most groups at present. Figure 1 gives a schematic overview. When mRNA electroporation is performed properly, transfection rates above 90% are feasible. This physical method requires no additional chemicals, which is beneficial under GMP conditions. Until now, most clinical trials applying mRNA-transfected DCs have used mRNAs encoding defined non-mutated tumor antigens. However, one can also perform sequencing of the tumor and identify mutated neo-antigens, which are considered more immunogenic, but are hampered by the fact that they seem to be rarely presented and constitute less than 1% of the HLA-ligandome [78,79,80]. To our knowledge, mRNAs encoding such mutated neo-antigens have only been used as a direct vaccine by injection into the lymph nodes [81], and thus far not in a DC vaccine.

An initial drawback of intracellularly expressed antigens is their limited presentation in MHC class II molecules for recognition by CD4^+^ helper T cells. While more experience was gathered in the field of DC vaccination, the relevance of T-cell-help in anti-tumor responses emerged. CD4^+^ helper T cells were shown to facilitate the generation of memory-type CD8^+^ T cells and, hence, this hurdle was overcome by fusing the mRNA-encoded antigens with targeting sequences, which direct the antigen towards the lysosomal compartment, thus mediating MHC class II-restricted presentation in addition to MHC class I-restricted presentation (Figure 1). Common targeting sequences were derived from lysosomal-associated membrane proteins (LAMP) like LAMP-1 [57] and DC-LAMP [82].

An additional merit of mRNA electroporation lies in the possibility of introducing not only antigens, but also functional proteins into the DCs, thus manipulating their phenotype and providing additional activation and maturation signals. The group around Kris Thielemans developed a DC-maturation process completely independent of exogenous cytokines by utilizing a mix of mRNAs encoding CD40L, CD70, and a constitutively active TLR4. This formulation was termed TriMix, and was used in a variety of clinical trials [40,41,42,43,46] (reviewed in [83]). Others used cytokine-matured DCs transfected with CD40L [84] to treat renal cell carcinoma [36] or HIV infection [72,73]. We commonly transfected cytokine-matured DC, but recently utilized a constitutively active form of IKKß to improve T-cell [85] and NK-cell activation [86], which will be tested in a phase I clinical trial soon.

### 3.4. The Total Tumor RNA Approach

The use of one or a few defined tumor antigens still limits the possibly induced anti-tumor immunity, and it has been shown that human anti-melanoma immunity is dominated by T-cell responses directed against somatically and individually mutated antigens [87]. Hence, the idea arose to use the entire tumor transcriptome by extraction, and, if needed, PCR-based amplification of tumor mRNA for subsequent electroporation into the DCs.

This allows an individualized treatment exploiting the complete antigenic repertoire of a given tumor, even if possible defined rejection antigens are yet unknown. In this aspect, total tumor mRNA is similar to the use of dead tumor cells but is not restricted by limitations regarding the size of excised tumors, reproducibility and validation. Exploitation of the total antigenic repertoire is considered critical as it targets not only overexpressed antigens but also the mutated proteins including both passenger as well as oncogenic driver mutations [88,89] and the emerging class of non-mutated neoantigens [90].

## 4. From the Bench to the Bedside

The potential clinical benefit of RNA-loaded DC therapy was first demonstrated in mouse tumor models. Mice treated with DCs pulsed with RNA from ovalbumin (OVA)-expressing tumor cells were protected against a challenge with OVA-expressing tumor cells [91]. In the same study, mice in the poorly immunogenic, highly metastatic, B16/F10 (B16) tumor model demonstrated a dramatic reduction in lung metastases in animals treated with DCs pulsed with tumor-derived RNA. Again using the B16 model, a second study showed that treatment with bone marrow-generated DCs, pulsed with either B16 cell extract or B16 total RNA induced specific CTLs against B16 tumor cells [92]. This treatment was able to protect animals from tumor located in the central nervous system (CNS), and led to prolonged survival in mice with tumors placed before initiation of therapy [92].

When this technology was taken from the bench to the bedside (as schematically represented in Figure 2), already the initial clinical trials were promising, demonstrating feasibility and immunogenicity, as well as hints for clinical efficacy. For example, a DC/RNA vaccine was explored in a phase I trial to treat eleven subjects presenting with metastatic RCC [31]. While the primary objectives of this study were safety, feasibility, and immunological assessment, it was noteworthy that tumor-related mortality was unexpectedly low among the 10 evaluable subjects who received the prescribed three administrations. The calculated mean survival following nephrectomy was 19.8 ± 3.1 months, although survival interpretation was confounded by the post-study therapies (predominately cytokine) received by most subjects. There were no adverse drug reactions with the exception of five subjects who experienced grade I injection site reactions consisting of inflammatory skin erythema lasting 48–72 h. A polyclonal tumor-specific T-cell response was detected in six subjects evaluable for immune response following DC treatment. Comparable results were observed in a phase I/II trial of melanoma therapy with autologous tumor mRNA [93,94]. The published clinical trials using tumor-RNA-transfected DCs are summarized in Table 1, Table 2 and Table 3.

Defined non-mutated antigens were used in most clinical trials with RNA-transfected dendritic cells and have proven immunogenic while autoimmunity was rarely observed, except for some cases of vitiligo in melanoma patients immunized with antigens expressed in both melanoma cells and melanocytes.

In the early studies with RNA-loaded dendritic cells, a simple co-incubation of DCs and RNA as opposed to electroporation was used. Since 2005, however, electroporation has commonly been used as a method to actively introduce mRNA into cytoplasm of the DCs. A total of 47 publications so far have described clinical phase I and II trials using RNA-loaded DCs to treat cancer and virus-infections. Within those, 781 treated patients were described. Most of those suffered from cutaneous melanoma (289) followed by viral infections (85), urogenital (79), and prostate cancer (73). For the complete summary, please see Table 1. In 10 of these trials, cells were loaded with autologous tumor-RNA, one used allogenic tumor RNA, and the other 36 used various defined antigens (Table 1).

The last publication in a clinical trial with RNA-loaded dendritic cells dates back to 2018. However, several active clinical trials with mRNA-transfected DCs to treat cancer are listed in clinicaltrials.gov. By November 2019, 18 active trials (Table 4) were listed mainly performed in the US (8), followed by Belgium (3), and Norway (3).

### 4.1. Clinical Efficacy

Within the large number of phase I/II DC trials that have been published, mixed responses (disappearance of some but not necessarily all metastases, also with appearance of new ones) and stabilization of disease were usually reported in a subset of patients. Objective responses, classically defined by disappearance of all tumors (CR) or a reduction of ≥50% (PR) were, however, less frequently observed. Interestingly, however, while overall response was found to be only 3.8% with non-DC-based cancer vaccines, in a much-debated article by Rosenberg et al., tumor regression was seen in 7.1% of patients receiving DC vaccination [95,96,97]. In select DC-vaccination trials, regressions were observed at higher rates, such as in DC-based vaccination for non-Hodgkin lymphoma targeting tumor-specific idiotype immunoglobulin (response rate of 31.6% [13,98]), or in melanoma trials when DCs were loaded with dying autologous tumor cells to vaccinate against the total antigenic repertoire of the individual tumors (20% overall response rate in stage IV melanomas [99,100]). These observations support the use of DC vaccines that target the antigenic repertoire of a given tumor as it can be achieved by loading DCs with total tumor mRNA as a technically more elegant approach which can be validated and thus be performed well under GMP conditions.

According to a review published by Ridgeway in 2003 [101], 78 of 98 analyzed trials included patient outcomes, although none of the clinical studies was designed to demonstrate the efficacy of the DC treatment. There was evidence of clinical response in at least one subject in 48 of the clinical trials, and one or more subjects experienced a complete response (CR) in 16 trials.

A review from Engell-Noerregaard et al. published in 2009 analyzed DC-based vaccination of patients with malignant melanoma [102]. A total of 38 articles were included for analysis, including 626 melanoma patients treated with DC-based vaccines. The objective response rate (CR and PR) was 9% with 20 (3%) complete responses and 37 (6%) partial responses. The clinical response rate (CR, PR, and SD) was 30% with 133 patients (21%) having stable disease. Apart from suggesting a clinical benefit in one third of the patients, the analysis was also interesting because it was found that SD was significantly associated with induction of antigen-specific T cells (*p* = 0.0003).

Regarding efficacy, it must, however, be emphasized that it has become clear that for active immunotherapies which lead to activation of tumor-specific T cells by either specific active vaccination (Dendreon’s first generation DC vaccine Provenge™ [103]) or antigen-unspecific immune activation by taking off the brake from the immune system (anti-CTL-A4 treatment with Ipilimumab™ [104,105]), prolonged overall survival does not necessarily require regressions as defined by classical response criteria. Researchers in the cancer vaccine field were the first to point to this possibility. Indeed, the Provenge™ DC vaccine phase III trials provided the first proof for this concept because time to progression was not significantly prolonged while OS was, so that finally this vaccine got approved by the FDA for the treatment of androgen-independent, metastatic prostate cancer. Treatment with the anti-CTL-A4 antibody Ipilimumab™ in phase II and the subsequently published phase III trial exhibited 4 response patterns associated with survival, with only two of them corresponding to classical regressions [104].

In retrospect, one has to state that it was very optimistic to expect DC vaccines or any other cancer vaccine (such as fashionable neo-antigen vaccines [106]) by themselves to frequently produce significant clinical benefit in the setting of established late stage malignancies like stage IV melanoma, given the increasing evidence that the tumor microenvironment dictates whether tumor-specific T-cell responses will successfully alter the course of the disease [107,108,109] as checkpoint molecules suppress spontaneously arising or vaccine-induced T cells. Nevertheless, the reported clinical responses with mRNA-transfected DCs are at least encouraging. Table 3 summarizes efficacy data from all published clinical trials using RNA-loaded DCs.

In the setting of stage IV melanoma, i.e., significant tumor-load, and also many other tumors, the use of DC vaccines, which are reliably immunogenic, is, therefore, now best explored in combination with other treatments such as anti-CTL-A4 and anti-PD1 treatment. In contrast, it is timely to test in randomized trials immunogenic DC vaccines alone in the setting of minimal tumor load as performed in the adjuvant treatment of resected monosomy 3 uveal melanoma patients (NCT01983748), because in this setting clinical benefit is a more realistic possibility.

### 4.2. Safety of DC Vaccine Therapy

In general, DC-based vaccination is well tolerated, and few severe side effects have been reported. The events most often reported after vaccination with antigen-loaded dendritic cells are local reactions at the dendritic cell injection sites, flu-like symptoms (fever, chills, headache, and myalgia) and fatigue. These immune-related symptoms are meanwhile considered to be reactogenicity to the vaccine, and are valued as a sign of the immunostimulatory effectiveness of the treatment. The local symptoms observed after subcutaneous or intradermal application are usually absent at onset but appear upon repetitive vaccination indicating accumulation of T cells at the injection site/draining lymph node, the systemic symptoms fatigue and increase of temperature probably being an effect of cytokines released.

In about 20% of our patients infused with standard DCs, we have ourselves observed grade 1 to 2 flu-like reactions (including fever up to 39.4 °C) and constitutional symptoms within the first 72 h after infusion. Such side effects resolved upon treatment with paracetamol (usually 1 g administered i.v. followed by oral application) within 8 h. This delayed reaction resembles a mild grade 1 CRS (cytokine release syndrome), which is caused by T-cell activation and mediated by the release of pro-inflammatory cytokines into the plasma such as IL-1, TNFα, and IL-6. Indeed, we detected an increase of these cytokines in the blood of those patients who were vaccinated with standard DC and developed transient fever from 6 h after DC vaccination onwards.

Severe autoimmune side effects/immune-related adverse events (IRAE)—as now often observed as toxic side effect of anti-CTL-A4 [105] and anti-PD1 therapy—have not been a safety issue in the context of DC vaccination, even in patients vaccinated for prolonged periods including patients with tumor regressions. The induction of severe autoimmunity-related side effects is theoretically possible with DC-based immunotherapy. In the case of DC vaccines, so far, the induction of autoantibodies without clinical symptoms has been observed occasionally. Induction of overt autoimmune diseases, with the exception of the occurrence of cosmetically troublesome, yet otherwise harmless vitiligo caused by the spotty destruction of skin melanocytes has, however, not been described.

The absence of autoimmune side effects is, however, not due to the fact that DCs were not sufficiently immunogenic. Cosmetically disturbing vitiligo resulting from destruction of melanocytes in the skin was regularly observed in a small number of patients after vaccination with DC loaded with melanocyte differentiation peptides; yet no other organ damage occurred. As expected, this side effect has also been observed with DCs loaded with mRNA including total mRNA. Out of 31 patients vaccinated with monocyte-derived DC loaded with autologous tumor-RNA, 1 patient developed vitiligo [37]. In one of our phase I trials with stage IV melanoma patients (NCT00126685), 1 out of 8 fully evaluable patients developed vitiligo after vaccination with DC loaded with autologous tumor RNA. Within another phase I/II trial, 9 patients out of 42 developed vitiligo after vaccination with DC, electroporated with MelanA, Mage-A3, and survivin mRNA.

What has been additionally observed in DC trials are laboratory abnormalities including positive anti-nuclear antibody tests [16,19,56,57,110,111,112,113,114,115,116], positive anti-dsDNA [111], positive anti-thyroid antibody tests [16,113,117,118], and positive rheumatoid factor [56,110,115,116,119]. Apart from four cases of thyroiditis [110], development of autoimmune antibodies was, however, not associated with clinically manifested autoimmune disorders such as systemic lupus erythematosus, rheumatoid arthritis, or dermatomyositis. In a trial using allogeneic DCs loaded with autologous renal cell carcinoma lysate, one patient experienced WHO grade IV thrombocytopenia [120], but it remained unclear whether thrombocytopenia was a side effect of the drug cyclophosphamide used in this trial too, from paraneoplastic origin, or triggered/aggravated by the allogeneic cell therapy.

Side effects observed in 46 publications describing the experience with DCs loaded with either defined RNA or RNA extracted from tumor cells (Table 4) did not significantly differ from those observed in the larger number of trials employing also monocyte-derived DCs but loaded by other methods but introduction of antigen mRNA (such as co-incubation with peptides or dead tumor cells). In the 781 patients treated with mRNA-loaded DCs, SAEs of more than grade II rarely occurred (Table 4). Often the attribution of such SAEs to the administered DCs has not been clear [57,58,69]. In one case, fatigue grade III occurred [38]. One study that must be mentioned here was recently described by Bol et al. [47]. They combined adjuvants from conventional preventive vaccines with DCs which resulted in heavy side effects including grade III flu-like symptoms, local reactions including purulent discharge and liver toxicity. All these symptoms were transient in nature and can be clearly attributed to the use of the adjuvants, because side effects of such extent have never been observed before, and were never observed afterwards.

Overall, the safety profile of DC vaccination including DCs transfected with mRNA is very good—notably as compared to any other treatment regimen for advanced malignancies.

### 4.3. Challenges and Future Perspectives of DC Vaccine Therapy

Although, as described above, DC vaccine therapy has clear merits, it is a very personalized medicinal product, requiring well-educated staff and a GMP-compliant facility for production, limiting its application. Depending on which source of mRNA is used for the transfection of the DCs, the costs and applicability can differ. For example, to gain a completely individualized product using autologous amplified total tumor RNA and autologous DCs, one has to obtain enough tumor RNA to perform the amplification procedure, and the RNA has to be produced for each patient. The same is true for the application in which individually mutated mRNAs are picked for the transfection, which additionally causes high costs for the sequencing of the tumor to find these mutations. On the other side, an off-the-shelf approach can also be chosen by using prepared mRNAs encoding non-mutated antigens often expressed in the tumor, which reduces the costs to some extent. Nonetheless, we strongly believe that the merits of mRNA-DC vaccine therapy overrule the above-mentioned disadvantages.

Adding DC vaccination to any of the standard therapies seems reasonable, since there is sound evidence that these standard regimens will enhance the T-cell induction by vaccination so that an enhanced clinical effect is possible, and a negative impact of the DC vaccine regarding clinical benefit of the standard therapy is unlikely. Importantly, there is no evidence for significant enhancement of undesired side effects as cancer vaccines including DC vaccines have been used without unexpected or clearly enhanced toxicity problems in man together with chemotherapy [121,122,123,124], as well as immune checkpoint blockade (ICB) [43,81,125,126,127,128]. With respect to combination with checkpoint blockade, anti-PD-1 is a preferred backbone for combinations with small molecules and immune stimulatory agents, but for resistant tumors (like uveal melanoma) double checkpoint blockade is used for triplet therapies as a novel concept. This is evidenced by a search in clinicaltrials.gov, which shows over 120 phase I triplet trials.

Among the 39 melanoma phase I trials employing double checkpoint blockade, there are combinations with HDACi, IDOi, etc., but 15/39 melanoma studies involve additional immune stimulatory drugs such as cytokines (hu14.18-IL2, NKTR-214 IL-2, IL-15) or vaccine-like stimulatory agonistic antibodies (anti-OX40, anti-GITR, anti-ICOS). Importantly, 2 of the 15 melanoma trials use vaccines as combination partner, specifically neo-antigen peptide vaccine plus Montanide (NCT03929029) and multiple class I peptides and Montanide ISA 51VG (NCT01176474). Another six trials in other tumor indications but melanoma also combine double ICB with vaccines. Four trials combining double ICB plus vaccines have already entered phase II (NCT03190265, NCT03639714, NCT02054520, and NCT03406715).

It is perhaps unexpected that triplet trials using vaccines as partners for toxic double checkpoint blockade (55% grade 3–4 irAE, 33% DLT) are that advanced. On the other hand, it is logical, as all the vaccine strategies mentioned above have a very low documented toxicity, and thus qualify as preferred combination partners compared to small molecules, cytokines or agonistic checkpoint molecules. This appears particularly true for DC vaccines. The Tri-Mix DC vaccine developed by K. Thielemans’ group was also given in combination with anti-CTL-A4, but again no grade 3 or 4 side effects occurred [43], even though Ipilimumab was used at the high dose level of 10 mg/kg, which because of the increased toxicity is avoided nowadays.

The combination of cancer vaccines (including DC vaccines) with chemotherapy has also been explored in the clinic (for a review also of ongoing trials see [129,130,131,132]). While significant synergism seems apparent only in few trials, most of them DC studies [133,134,135,136], a negative impact has never been reported, as the rules for a successful combination—also derived from animal studies—have been taken into account. These principles are: (1) avoid high-dose chemotherapy, (2) avoid combination after prolonged chemotherapy which results in general immunosuppression, and (3) avoid the concomitant administration of vaccines and cytotoxic drugs but rather administer about 1–2 weeks later. This circumvents the inhibition of activated, proliferating vaccine-induced T cells, and can dramatically foster T-cell responses by depletion of unwanted myeloid cells [133]. Gemcitabine has been explored in mice and humans in combination DC and other cancer vaccines with promising results [137,138,139,140,141]. Reassuringly, gemcitabine also synergizes with two other types of immune therapy, namely oncolytic virotherapy and ICB [142,143,144]. Like gemcitabine, fotemustine has already been tested in combination with prolonged vaccination with promising results and no added toxicity [145]. Interestingly, the combination with other immunostimulatory agents, namely IFN alpha + IL-2 and anti-CTL-A4 ICB, was also promising and clinical activity was observed again without evidence for enhanced side effects [146,147].

## 5. Conclusions

Undoubtedly, DC-based cancer vaccines are safe and feasible, and RNA-transfection is emerging as an ideal method for antigen-loading and functional manipulation of the applied cells. While other new cancer treatment regimens involve serious side effects, DC vaccination rarely produces adverse events higher than grade II. This allows a combination treatment in patients with a high tumor burden where DC-based monotherapy yielded only limited clinical results. Additionally, DC vaccines should be further extended to the adjuvant setting, to circumvent the massive immunosuppression exerted by a late stage tumor. Exploration of alternative DC origins and maturation protocols and the functional manipulation of the DCs by transfection with mRNA encoding proteins that trigger activation pathways is a consequent perpetuation to increase their immunogenicity. Currently running and future clinical trials explore these new approaches.

## Figures and Tables

**Figure 1 pharmaceutics-12-00092-f001:**
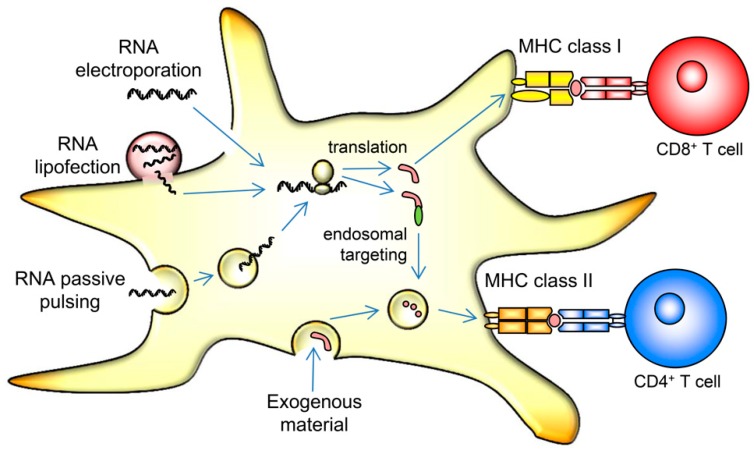
Schematic representation of an mRNA-transfected DC. Endocytosed and phagocytosed material is usually presented in MHC class II to CD4^+^ helper T cells and can be cross-presented in MHC class I only under certain circumstances (not depicted). Proteins from the cytoplasm are, in contrast, primarily presented in MHC class I. mRNA can be introduced into cells by passive pulsing, relying on intrinsic and yet unknown means of uptake. It can be complexed with lipid reagents that mediate entry into the cytoplasm, or it can be transfected by electroporation. By applying a short electric pulse, pores in the cell membrane temporarily open, allowing entry of the RNA molecules. The transferred mRNA is then translated in the cytoplasm, and the encoded antigens are hence presented in a MHC class I context. By encoding signaling and targeting sequences (green) fused to the antigenic protein, this can also be directed towards the endosomal pathway, resulting in efficient MHC class II presentation.

**Figure 2 pharmaceutics-12-00092-f002:**
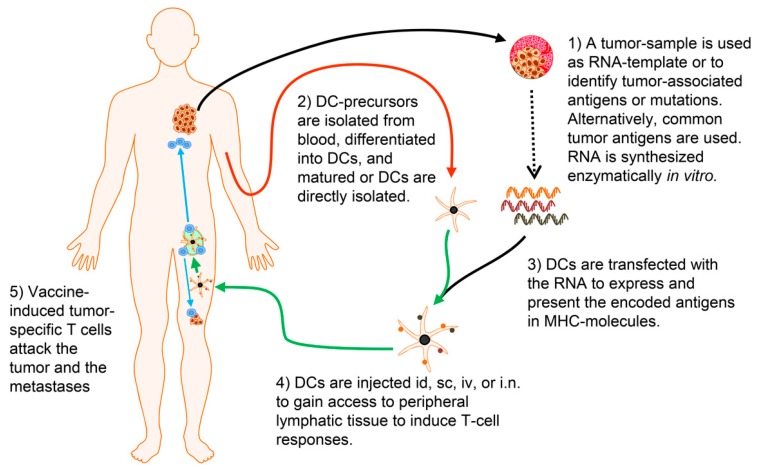
Concept of therapeutic vaccination with mRNA-transfected DCs. (**1**) Tumor material is isolated by surgery or biopsy. From this material, mRNA can be directly isolated. This RNA is usually amplified via a PCR-based method to gain sufficient amounts of mRNA. The tumor material can also be analyzed by sequencing, immunohistology, or other methods to identify antigens associated with this tumor, including somatic mutations. New bioinformatical methods can be used to identify the most promising T-cell epitopes for each individual patient. mRNA molecules that encode these antigens are transcribed in vitro from plasmid templates. (**2**) Cells (usually monocytes) that can be differentiated into DCs in vitro are isolated from the patient’s blood. Alternatively, DCs, which are present in the blood, can be used. These DCs are matured to become immunogenic. (**3**) DC are transfected with the RNA to express the encoded tumor antigens. The DCs’ own processing machinery degrades and presents the included T-cell epitopes in MHC. (**4**) The DCs are injected into the patient. Intradermal (id) and subcutaneous (sc) injection require migration via the lymphatic vessels towards the draining lymph node. Intravenous (iv) injection necessitates the transfer from the blood stream into lymphatic tissue. The direct injection into lymph nodes (i.n.) is an elegant approach, but is technically very difficult. (**5**) If the vaccine is successful, the DCs present the tumor-specific epitopes to T cells which are activated and attack the malignant tissue. Usually the DCs are injected repetitively to boost and maintain the responses. (The Motifolio Scientific Illustration Toolkit was used for the generation of this figure).

**Table 1 pharmaceutics-12-00092-t001:** Antigens and diseases treated with RNA-transfected DCs.

RNA Used	Malignancy/Disease	Patients Vaccinated	Phase	DC Culturing	Maturation	Reference
Autologous tumor RNA (aT-RNA)	Colorectal ca m	15	I	Standard (including FCS)	No	[29]
	Colon ca	1	0	Standard	TNFα	[30]
	Renal cell ca m	10	I	Standard	No	[31]
	Pediatric brain tumors	7	I	Standard	No	[32]
	Pediatric neuroblastoma stage IV	8	I	Standard	No	[33]
	Renal cell ca m, ovarian ca	11	ns	Standard	MCMM	[34]
	Melanoma stage IV	6	I	Standard (Clinimacs)	TNFα + PGE_2_	[35]
	Renal cell ca m	21	II	Standard	TNFα + PGE_2_ + IFNγ + CD40L-mRNA	[36]
	Melanoma stage IV	31	I/II	Standard	MCMM	[37]
aTSC-RNA (tumor stem cells)	Glioblastoma	7	I/II	Standard (5 days)	MCMM	[38]
Allogeneic tumor RNA (3 human cancer cell lines)	Prostate ca	19	I/II	Standard	MCMM	[39]
MAGE-A1- MAGE-A3-, MAGE-C2-, MelanA-, tyrosinase-, and/or gp100-DC-Lamp mRNA	Melanoma stage III and IV	30	I *	Standard (6 days)	TriMix or polyIC + CD40L-mRNA	[40]
MAGE-A3-, MAGE-C2-, tyrosinase-, gp100-DC-Lamp mRNA	Melanoma stage IIIC and IV	35	I *	Standard (6 days)	TriMix- mRNA	[41]
	Melanoma stage IIIC and IV	15	IB	Standard (6 days)	TriMix-mRNA	[42]
	Melanoma stage IIIC and IV	39	II	Standard (6 days)	TriMix-mRNA	[43]
gp100 or tyrosinase mRNA	Melanoma m stage III	11	I/II	Standard	MCM + TNFα + PGE_2_	[44]
gp100 and tyrosinase mRNA	Melanoma stage III and IV	45	I/II	Standard (5–7 days)	MCM + PGE_2_ + TNFα	[45]
	Melanoma stage III and IV	15	I/II	Standard	TriMix-mRNA	[46]
	Melanoma stage III and IV	28	I/II	Standard	TLR-agonists from conventional vaccines	[47]
	Uveal melanoma	23	I/II	Standard	ns	[48]
MelanA, MAGE-A3, gp100, and tyrosinase mRNA	Melanoma m	12	I	Standard (5 days)	MCMM	[49]
hTERT, survivin, and p53 mRNA	Advanced melanoma	22	I	Standard	ns	[50]
MAGE-A3, survivin, BCMA mRNA	Multiple myeloma stage II/III	12	I	Standard	MCMM	[51]
MUC1 and survivin mRNA	Renal cell carcinoma	28	I/II	Standard (4 days)	TNFα	[52]
CEA mRNA	Pancreatic cancer	3	ns	Standard	No	[53]
	CEA expressing malignancies	37	I/II	Standard	No	[54]
	Colorectal cancer m	5	I/II	Standard	MCMM	[55]
PSA mRNA	Prostate cancer m	13	I	Standard	No	[56]
hTERT mRNA, +/− LAMP	Prostate cancer m	20	I	Standard	MCMM	[57]
	AML	21	II	ns	ns	[58]
PSA, PAP, survivin, and hTERT mRNA	Castration-resistant prostate ca	21	II	Standard	ns	[59]
Folat receptor mRNA	Ovarian cancer m	1	0	Standard	MCMM	[60]
WT1 mRNA	AML	10	I/II	Standard (6 days)	TNFα+ PGE_2_	[61]
	Uterine cancer	6	I/II	Standard (6 days)	TNFα + IL1ß	[62]
	Ovarian	2	I/II	Standard (6 days)	TNFα + IL1ß	[63]
WT1 mRNA +/− DC-Lamp	AML	20	II	Standard (6 days)	TNFα+ PGE_2_	[64]
MUC1 mRNA	Pancreatic cancer	42	ns	Standard (6 days)	TNFα	[65]
CMV pp65.LAMP mRNA	Glioblastoma	12	I	Standard	MCMM	[66]
HSP70	HCV-related hepatocarcinoma	12	I	Standard	TNFα	[67]
CMV pp65 mRNA	Hematopoietic stem cell transplantation	7	ns	Standard (6 days clinimacs)	TNFα, PGE_2_	[68]
	Glioblastoma	11	I	Standard from CD34+	ns	[69]
	Glioblastoma	9	I	Standard	MCMM	[70]
Gag and Nef mRNA	HIV infection	10	I/II	Standard (5 days)	MCMM	[71]
Gag, Vpr, Rev and Nef mRNA	HIV infection	10	I/II	Standard	TNFα + IFNγ+ PGE_2_ + CD40L mRNA	[72]
	HIV infection	35	IIB	Standard	TNFa + IFNγ + PGE_2_ + CD40L-mRNA	[73]
Gag-, Tat-, Rev-, and Nef-DC-Lamp mRNA	HIV infection	17	I/IIa	Standard (6 days)	MCMM	[74]
Gag- and Tat-Nef-Rev-DC-Lamp mRNA	HIV infection	6	I/II	Standard (clinimacs)	TNFα + PGE_2_	[75]

m: metastatic, ca: cancer, aT-RNA: autologous tumor RNA, aTSC-RNA: autologous tumor stem cell RNA, AML: acute myeloid leukemia, MAGE: melanoma-associated antigen, Lamp: lysosome-associated membrane protein, hTERT: human telomerase reverse transcriptase, BCMA: B-cell maturation antigen, MUC1: mucin 1, CEA: carcinoembryonic antigen, PSA: prostate-specific antigen, PAP: prostatic acid phosphatase, WT1: Wilms Tumor 1, CMV: cytomegalovirus, HSP70: heat-shock protein 70, Gag: HIV group-specific antigen, Vpr: HIV viral protein R, Rev: HIV reverse transcriptase, Nef: HIV negative regulatory factor, Tat: HIV trans-activator of transcription, MCM: monocyte-conditioned medium, MCMM = MCM-mimic (TNFα, IL-1, IL-6, PGE_2_), Standard = Monocytes cultured in GM-CSF and IL-4 (default = 7 days), ns: not specified, * Specified as: “single center pilot clinical trial”.

**Table 2 pharmaceutics-12-00092-t002:** Clinical efficacy of RNA-based DC trials.

Study Reference	# Pts	Disease and Combination Treatment	DC Culturing	Maturation	RNA-Transfection	Clinical Response
[35]	6	Melanoma, stage IV	Standard (Clinimacs)	TNFα + PGE_2_	EP with aT-RNA	No objective clinical response
[44]	11	Melanoma stage III	Standard	MCM + TNFα + PGE_2_	EP with gp100 or tyrosinase mRNA	No information regarding clinical responses, vaccine-related CTL-responses in 7 pts
[41]	35	Melanoma, m	Standard (6 days)	TriMix- mRNA	EP with MAGE-A3, -C2, tyrosinase, gp100 mRNA	Of 20 pts with measurable disease 11 pts SD, 9 pts PD. Of 15 pts without measurable disease 9 showed relapse.
[45]	45	Melanoma stage III and IV, m	Standard (5–7 days)	MCM + PGE_2_ + TNFα	EP with gp100 and tyrosinase mRNA	Stage III: median PFS 34.3 months; mOS not reached. Stage IV (19 pts): 6 SD, 1 PR, 12 PD; mOS 24.1 months (patients with positive immunomonitoring)
[49]	12	Melanoma, m	Standard (5 days)	MCMM	EP with gp100, MelanA, tyrosinase, and MAGE-A3 mRNA +/− IP siRNA	1 pt PR 1 pt CR mOS 35 months
[42]	15	Melanoma	Standard (6 days)	TriMix-mRNA	EP with gp100-, tyrosinase-, MAGE-A3-, and -C2-DC-Lamp mRNA	2 pts with CR 2 pts with PR 4 pts with SD
[46]	15	Melanoma	Standard	TriMix-mRNA	EP with gp100 and tyrosinase mRNA	Mpfs = 15.14 months mOS = 23.36 months 1 pt = not evaluable 7 pts with PD 2 pts with SD 1 pt with MR 3 pts with no evidence of disease
[40]	30	Melanoma (adjuvant)	Standard (6 days)	TriMix or polyIC + CD40L-mRNA	EP with MAGE-A1-, -A3-, -C2-, tyrosinase-, melanA-, and gp100-DC-Lamp RNA	mRFS = 22 months St IIIB/C = 18 months, OS = not reached St III = 36 months; OS = 6.2 years St IIB IIC II 24–27 months; OS = 5.3 years mOS = not reached
[47]	28	Melanoma stage III and IV	Standard	TLR-agonists from conventional vaccines	EP with gp100 and tyrosinase mRNA	4 pts with SD
[37]	31	Advanced melanoma	Standard	MCMM	EP with aT-RNA	1 pt with PR 3 pts with SD OS 10 months
[50]	22	Malignant melanoma Cyclophosphamide	Standard	ns	EP with hTERT, survivin, p53 mRNA	9 pts with SD mPFS 3.1 months mOS 10.4 months
[43]	39	Pretreated advanced melanoma Ipilimumab	Standard (6 days)	TriMix-mRNA	EP with MAGE-A3-, -C2-, tyrosinase-, and gp100-DC-LAMP mRNA	8 pts with CR 7 pts with PR 6 pts with SD mPFS 27 weeks mOS 59 weeks
[48]	23	Uveal melanoma	Standard	ns	EP with gp100 and tyrosinase mRNA	mDFS 34.5 months mOS 51.8 months
[60]	1	Advanced serous papillary ovarian cancer stage IIIc	Standard	MCMM	EP with folatR mRNA	1 pt PR
[62]	2	Ovarian cancer	Standard (6 days)	TNFα + IL1ß	EP with WT1 mRNA	Patients with ovarian carcinosarcoma showed OS of 70 months (vs 15.5 months in historical controls).
[63]	6	Uterine cancer	Standard (6 days)	TNFα + IL1ß	EP with WT1 mRNA	OS of 10 to 11 months compared to 2–5 months historical controls
[31]	10	Renal cell carcinoma, stage III or IV	Standard	No	co-incubation with aT-RNA	7 pts SD/slow progression
[34]	11	Renal cell cancer, m (10 pts), ovarial carcinoma (1pt) Ontak^®^	Standard	MCMM	EP with aT-RNA	Increase in tumor-specific CTL, no information on clinical responses
[52]	28	Renal cell cancer cytokine-induced killer cells	Standard (4 days)	TNFα	EP with MUC-1 and survivin mRNA	4 pts with CR: 2 > 10 months; 2 > 15 months 7 pts with PR (6–21 months) 10 pts with SD (5–21 months) 6 pts with PD/1 death
[36]	21	Renal cell cancer sunitinib	Standard	TNFα + PGE_2_ + IFNγ + CD40L-mRNA	EP with aT-RNA	5 pts with PR 8 pts with SD 13 pts with PR + SD 8 pts with PD Median OS:30.2 months
[56]	13	Prostate cancer, m	Standard	No	co-incubation with PSA mRNA	1 pt decrease of PSA level, 5 pts reduction PSA log slope, 3 pts transient elimination of tumor cells in peripheral blood
[39]	19	Prostate cancer, androgen resistant	Standard	MCMM	EP with allogeneic tumor RNA (3 human cancer cell lines)	11 pts SD (PSA) 13 pts decreased log slope PSA
[57]	20	Prostate cancer, m	Standard	MCMM	EP with hTERT mRNA +/− LAMP	No objective clinical response increase in hTERT-specific CTL and molecular clearence of circulating micrometastases
[59]	21	Castration-resistant prostate cancer docetaxel	Standard	ns	EP with PSA, PAP, survivin, hTERT mRNA	mPFS 5.5 months
[32]	7	Pediatric brain tumors	Standard	No	co-incubation with aT-RNA	0 pt CR, 1 pt PR, 2 pts SD
[33]	8	Pediatric neuroblastoma stage IV	Standard	No	co-incubation with aT-RNA	No objective clinical response
[38]	7	Glioblastoma	Standard (5 days)	MCMM	EP with aT-RNA	Median PFS of 694 days vs. 236 days in historical controls Median OS of 759 days vs. 585 days in historical controls
[66]	12	Glioblastoma injection site preconditioned with tetanus toxoid	Standard	MCMM	EP with CMV pp65 mRNA	mPFS of 10.8 months; mOS 18.5 months
[69]	11	Glioblastoma temozolimide DCs mixed with GM-CSF	Standard from CD34+	ns	EP with CMV pp65 mRNA	mPFS 25.3 months mOS 41.1 months
[70]	9	Glioblastoma adoptive T-cell transfer	Standard	MCMM	EP with CMV pp65 mRNA	increase in polyfunctinal pp65-specific T cells
[53]	3	Pancreatic adenocarcinoma, CEA expressing	Standard	No	co-incubation with CEA mRNA	3 pts SD
[65]	42	Pancreatic cancer cytotoxic lymphocytes gemcitabine	Standard (6 days)	TNFα	EP with MUC-1 mRNA	1 pt with CR, 3 pts with PR, 22 pts with SD16 pts with PD mOS 13.9 months 1-year survival rate 51.1%
[54]	37	CEA expressing cancer, m (24 tumor bearing, 13 tumor free)	Standard	No	co-incubation with CEA mRNA	1 pt CR, 2 pts PR, 2 pts SD
[29]	15	Colorectal cancer, m	Standard (including FCS)	No	co-incubation with aT-RNA	No objective clinical response
[55]	5	Colorectal cancer, m	Standard	MCMM	EP with CEA mRNA	Median progression free survival of 26 months
[30]	1	Adenocarcinoma, m	Standard	TNFα	lipofection of aT-RNA	No objective clinical response
[51]	12	Multiple myeloma	Standard	MCMM	EP with BCMA, MAGE3, and survivin mRNA	After 25 months 10 of 12 pts still alive with 5 pts having SD, 5 pts having PD
[58]	21	AML	ns	ns	EP with hTERT mRNA, +/− LAMP	“vaccination with hTERT-DCs may be associated with favorable recurrence-free survival”
[64]	30	AML	Standard (6 days)	TNFα+ PGE_2_	EP with WT1 mRNA +/− DC-lamp	9 pts with molecular remission 4 pts with SD relapse reduction rate of 25%
[68]	7	4 healthy volunteers, 3 HSCT recipients	Standard (6 days clinimacs)	TNFα, PGE_2_	EP with CMV pp65 mRNA	No survival data (vaccination to induce CMV cellular response)
[67]	12	HCV-related hepato-carcinoma	Standard	TNFα	EP with HSP70 mRNA	2 pts with CR (min. 33 and 44 months)
[72]	10	HIV infection	Standard	TNFα + IFNγ+ PGE_2_ + CD40L mRNA	EP with Gag, Vpr, Rev, and Nef mRNA	7 pts HIV-specific proliferative immune response
[74]	17	HIV infection	Standard (6 days)	MCMM	EP with Tat-, Rev-, or Nef-DC-Lamp mRNA	Vaccine-specific immune response
[75]	6	HIV infection	Standard (clinimacs)	TNFα + PGE_2_	EP with Gag-DC-Lamp or Tat-Rev-Nef-DC-Lamp mRNA	Vaccine-specific immune response
[71]	10	HIV infection	Standard (5 days)	MCMM	EP with Gag and Nef mRNA	increased but short-lived CD4-responses against HIV gag and nef
[73]	35	HIV infection	Standard	TNFa + IFNγ + PGE_2_ + CD40L-mRNA	EP with Gag, Vpr, Rev, and Nef mRNA	none

m: metastatic, pt(s): patient(s), ns: not specified, St.: stage, EP: electroporation, IP: immunoproteasome, (m)OS: (median) overall survival, (m)RFS: (median) relapse free survival; (m)PFS: (median) progression free survival, CR: complete response, PR: partial response, SD: stable disease, PD: progressive disease, HSCT: haematopoietic stem cell transplantation, MCM: monocyte-conditioned medium, MCMM = MCM-mimic (TNFα, IL-1, IL-6, PGE_2_), aT-RNA: autologous tumor RNA, aTSC-RNA: autologous tumor stem cell RNA, AML: acute myeloid leukemia, MAGE: melanoma-associated antigen, Lamp: lysosome-associated membrane protein, hTERT: human telomerase reverse transcriptase, BCMA: B-cell maturation antigen, MUC1: mucin 1, CEA: carcinoembryonic antigen, PSA: prostate-specific antigen, PAP: prostatic acid phosphatase, WT1: Wilms Tumor 1, CMV: cytomegalovirus, HSP70: heat-shock protein 70, Gag: HIV group-specific antigen, Vpr: HIV viral protein R, Rev: HIV reverse transcriptase, Nef: HIV negative regulatory factor, Tat: HIV trans-activator of transcription, ns: not specified. Standard = Monocytes cultured in GM-CSF and IL-4 (default = 7 days).

**Table 3 pharmaceutics-12-00092-t003:** Adverse events in trials using DCs loaded with mRNA.

Study (Reference)	# Pts	Disease + Combination Treatment	Transfection	Route and Target Dose	Safety Summary
[35]	6	Melanoma, stage IV	EP with aT-RNA	5 × 10^6^ sc in 3-weekly intervals for 4 cycles	2 pts with fatigue (1 grade I, 1 grade II), 2 pts with nausea (1 grade I, 1 grade II), 1 pt with anorexia (grade II), 1 pt with arthralgia (grade I), 1 pt with confusion (grade I), 2 pts with diarrhea (grade I), 1 pt with hemorrhage (grade I), 1 pt with local reaction (grade I), 1 pt with myalgia (grade II), 1 pt with abdominal pain (grade II), 1 pt with bone pain (grade I), 1 pt with speech disorder (grade I), 1 pt with vomiting (grade I), 1 pt with wound infection (grade I)
[44]	11	Melanoma stage III	EP with gp100 or tyrosinase mRNA	1.5 × 10^7^ in biweekly intervals for 3 cycles	No side effects described
[41]	35	Melanoma, m	EP with MAGE-A3, -C2, tyrosinase, gp100 mRNA	4.3 × 10^7^ id 4 times in biweekly intervals; further vaccinations in case of residual vaccine after an 8 week interval	all pts: local reaction (grade II) 2 pts fever, myalgia, and asthenia grade II
[45]	45	Melanoma, m	EP with gp100 and tyrosinase mRNA	12 × 10^6^ cells 3 id times in biweekly intervals; 2 maintenance cycles for stable patients after 6 months respectively	local reaction: 23 pts grade I, 1 pt grade II flu like symptoms: 20 pts grade I, 10 pts grade II
[49]	12	Melanoma, m	EP with gp100, MelanA, tyrosinase, and MAGE-A3 mRNA +/− IP siRNA	10^7^ cells id 6 times in weekly intervals	No adverse events observed
[42]	15	Melanoma	EP with gp100-, tyrosinase-, MAGE-A3-, and -C2-DC-Lamp mRNA	Cohort 1: 2 × 10^7^ id, 4 × 10^6^ iv Cohort 2: 12 × 10^6^ id, 12 × 10^6^ iv Cohort 3: 4 × 10^6^ id, 2 × 10^7^ iv Cohort 4: 24 × 10^6^ iv 4 vaccinations in biweekly intervals, 5th vaccination with 10 weeks interval	11 pts local reaction grade II 3 pts chills grade II 8 pts flu like symptoms grade II 3 pts fever grade II
[46]	15	Melanoma	EP with tyrosinase and gp100 RNA	Up to 15 × 10^6^ cells i.n. 3 times with maintenance cycles every 6 months	4 pts local reaction grade I 4 pts flu like symptoms grade I
[40]	30	Melanoma (adjuvant)	EP with MAGE-A1-, -A3-, -C2-, tyrosinase-, MelanA-, and gp100-DC-Lamp RNA	~24 × 10^6^ id 4 to 6 times in biweekly intervals	30 pts local reaction grade II 1 pt fever grade II 1 pt flu like symptoms 7 pts vitiligo
[47]	28	Melanoma stage III and IV	EP with gp100 and tyrosinase mRNA	16 pts: 75 × 10^5^ to 3 × 10^7^ iv (2/3) and id (1/3) 12 pts: 15 × 10^5^ to 16 × 10^7^ intranodally 3 biweekly vaccinations per cycle, max 2 cycles in 6 months	flu-like: 11 pts grade I, 16 pts grade II, 1 pt grade III local reactions: 12 pts grade I, 13 pts grade II Hepatotoxicity: 9 pts grade I, 10 pts grade II, 5 pts grade III pneumonitis: 8 pts vitiligo 1 pt
[37]	31	Advanced melanoma	EP with aT-RNA	4 weekly injection 2 × 10^7^ intranodally (21) or id (10) then one id. 9 intranodally injected patients received IL-2	Mild flu-like symptoms in some pts, pain in tumor, inflammatory reaction at injection site (grade I and II) 1 pt: vitiligo grade I no long term toxicity
[50]	22	Malignant melanomacyclophosphamide	EP with hTERT, survivin, p53 mRNA	5 × 10^6^ intermitting with cyclophosphamide for 6 cycles	Grade III: 1 pt: lung embolus from leukapheresis-catherization Grade I and II: 13 pts fatigue, 12 pts nausea, 7 pts diarrhea, 5 pts anemia, 1 pt neutropenia, 1 pt: hyperthyroidism, 1 pt vitiligo, 1 pt myalgia. (all not attributed to either vaccine or cyclophosphamide)
[43]	39	Pretreated advanced melanoma Ipilimumab	EP with MAGE-A3-, -C2-, tyrosinase-, and gp100-DC-LAMP mRNA	4 × 10^6^ id and 2 × 10^7^ iv 1 h after Ipilimumab first 18 patients received one does DCs 2 weeks before Ipilimumab	DC-related: all pts: grade II injection site reactions 15 pts: grade I+II post-infusion chills 33 pts: grade I+II flu-like symptoms ICB-related: 14 pts: grade III+IV
[48]	23	Uveal melanoma	EP with gp100 and tyrosinase mRNA	up to 3 cycles of 3 biweekly iv and id injections in 6-month intervals	21 pts: grade I and II flu-like symptoms, 20 pts: grade I and II local reactions, 1 pt vitiligo
[60]	1	Papillary ovarian cancer stage IIIc	EP with folat-R-mRNA	2 to 50 × 10^6^ id in monthly intervals for 10 cycles	No side effects
[62]	2	Ovarian cancer	EP with WT1 mRNA	7–61 × 10^6^ cells id 4 times in weekly intervals in Imiquimod pretreated skin	No signs of toxicity
[63]	6	Uterine cancer	EP with WT1 mRNA	6–32 × 10^6^ cells id 4 times in weekly intervals Imiquimod pretreated skin	6 pts local reaction grade I
[31]	10	Stage III or IV renal cell carcinoma after nephrectomy	No EP, co-incubation with aT-RNA	8 pts: 10^7^ iv + 10^7^ id every 2 weeks for 3 cycles 2 pts: 3 × 10^7^ iv + 10^7^ id every 2 weeks for 3 cycles	5 pts with local reaction (grade I), 1 pt with anemia, 2 pts with dyspnea (both grade I, both considered unrelated to vaccine);
[34]	11	Renal cell cancer, m (10 pts), ovarial carcinoma (1pt) Ontak^®^ (7 pts)	EP with aT-RNA	10^7^ id at biweekly intervals for 3 cycles	4 pts with grade 1 rise of temperature and malaise (after Ontak^®^) 1 pt with elevation of RF (after Ontak^®^) 1 pt with transient ALT elevation (after Ontak)
[52]	28	Renal cell cancer cytokine-induced killer cells	EP with MUC-1 and Survivin mRNA	2 × 10^7^ to 5 × 10^7^ cells sc 4 times in 2 days intervals	Flu like symptoms and fever grade I and II
[36]	21	Renal cell cancer sunitinib	EP with aT-RNA	14 × 10^6^ cells	Vaccine-related - all grade I or II: 7 pts: injection site erythema, 5 pts: Injection site induration, 4 pts rash, 3 pts diarrhea, 3 pts fatigue, 2 pts nausea, 2 pts headache, 1 pt decreased weight, 1 pt hypertension, 1 pt dysgeusia
[56]	13	Prostate cancer, m	No EP, co-incubation with PSA mRNA	3 pts: 10^7^ iv + 10^7^ id for 3 cycles 3 pts: 3 × 10^7^ iv + 10^7^ id for 3 cycles 7 pts: 5 × 10^7^ iv + 10^7^ id for 3 cycles 2 week intervals	4 pts with local reaction (grade I) 4 pts with grade I fever accompanied by flu-like symptoms following injection 1 pt with transiently elevated ANA and RF
[39]	19	Prostate cancer, androgen resistant	EP with allogeneic tumor RNA (3 human cancer cell lines)	2 × 10^7^ either intranodally (10 pts) or id (9 pts) weekly for 4 cycles	No grade II to IV side effects. Erythema at injection sites, increased size of draining lymph nodes, minor pain at injection site or small increase in hot flushes.
[57]	20	prostate cancer, m	EP with hTERT mRNA +/− LAMP	10^7^ id in weekly intervals (3 or 6 cycles)	4 pts with constitutional symptoms (grade I) like fatigue or flu-like symptoms 18 pts with local reaction (grade I) 2 pts with transient elevation of ANA 1 pt with anemia and thrombocytopenia (grade III) considered unrelated to therapy
[59]	21	Castration- resistant prostate cancer Docetaxel	EP with PSA, PAP, survivin, hTERT mRNA	5 × 10^6^ twice during four Docetaxel-cycles, then one for 6 cycles, then only DCs every 3 months at patient decision	DC-related: local rash and pain only one pulmonary embolism related to leukapheresis procedure
[32]	7	Pediatric brain tumors	No EP, co-incubation with aT-RNA	5 × 10^6^/m^2^ iv + 5 × 10^6^/m^2^ id	No measurable toxicity, no signs of autoimmunity
[33]	8	Pediatric neuroblastoma stage IV	No EP, co-incubation with aT-RNA	5 × 10^6^/m^2^ iv + 5 × 10^6^/m^2^ id	No measurable toxicity, no signs of autoimmunity 1 pt with grade 1 skin reaction
[38]	7	Glioblastoma	EP with aT-RNA	10^7^ cells id; 2 vaccinations within first week, followed by 3 vaccinations in weekly intervals; rest of vaccinations in monthly intervals	fatigue: 6 pts grade I, 1 pt grade III 5 pts nausea/anorexia grade I pain: 3 pts grade 1, 1 pt grade II 1 pt constipation grade I
[66]	12	Glioblastoma injection site preconditioned with tetanus toxoid	EP with CMV pp65 mRNA	2 × 10^7^ cells id 3 times in biweekly intervals followed by monthly intervals	None
[69]	11	Glioblastoma temozolimide DCs mixed with GM-CSF	EP with CMV pp65 mRNA	three times 2 × 10^7^ in biweekly intervals then monthly 6 to 12 times into the groin	No AEs in response to DCs, but one grade III SAE in response to the co-injected GM-CSF
[70]	9	Glioblastoma adoptive T-cell transfer	EP with CMV pp65 mRNA	three times 2 × 10^7^ in biweekly intervals iv	2 pts reduced CD4 count (grade II), 1 pt reduced platelet count (grade I) 1 pt reduced Neutrophil and WBC count (grade II) 1 pt reduced hematocrit (grade I)
[53]	3	Pancreatic adenocarcinoma CEA expressing	No EP, co-incubation with CEA mRNA	10^7^ loaded and 10^7^ unloaded DCs id monthly for 6 cycles	1 pt with liver abscess, 1 pt with upper respiratory infection (both considered unrelated to vaccine)
[65]	42	Pancreatic cancer cytotoxic lymphocytes gemcitabine	EP with MUC-1 mRNA	4 × 10^5^ to 39 × 10^6^ cells id in monthly intervals	several grade 3 and adverse events, but attributed to T-cell transfer
[54]	37	CEA expressing cancer m (24 tumor bearing, 13 tumor free)	No EP, co-incubation with mRNA encoding CEA	11 pts: 10^7^ iv weekly for 4 weeks 4 pts: 3 × 10^7^ iv + 10^6^ id every 2 weeks for 4 cycles 14 pts: 10^8^ iv + 10^6^ id every 2 weeks for 4 cycles 8 patients additionally received 1.2 × 10^6^ units IL-2 s.c. group 2: 13 pts: 3 × 10^7^ iv + 10^6^ id every 2 weeks for 4 cycles	No acute toxicities (no evidence of anaphylactic reactions or other cardiopulmonary compromise) Rise of body temperature of 0.28 °C (0.5 °F) Rise of mean arterial pressure of 6mm Hg Unrelated or tumor-related: 1 pt with rise in hepatic transaminases (from grade I to grade III) 1 pt with myelodysplastic syndrome 6 months after completing therapy 1 pt with an upper extremity deep vein thrombosis
[29]	15	Colorectal cancer, m	co-incubation aT-RNA	4 × 10^6^ iv every 4 weeks for 4 cycles	2 pts with transient rigor and malaise
[55]	5	Colorectal cancer, m	EP with CEA mRNA	5 × 10^6^ id, 1.1 × 10^7^ iv on day 0, 7 and 15. 3 cycles.	Flu like symptoms grade I, fever grade I, local reaction grade I
[30]	1	Adenocarcinoma, m	No EP, lipofection of aT-RNA	3 × 10^7^ iv + 10^6^ id every 4 weeks for 4 cycles	No toxicities observed
[51]	12	Multiple myeloma	EP with BCMA, MAGE3, and survivin mRNA	15 × 10^6^ cells iv and 8 × 10^6^ cells id 3 times in biweekly intervals	8 pts local reaction grade I 10 pts fever, chills, malaise, muscle pain grade I/II
[58]	21	AML	EP with hTERT mRNA, +/− LAMP	3 to 32 vaccinations with 10^7^ DCs, first 6x in weekly intervals later biweekly	1 pt idiopathic thrombocytopenia purpura (grade III) no other severe toxicities reported
[64]	30	AML	EP with WT1 mRNA +/− DC-Lamp	5 × 10^6^, 10^7^ or 2 × 10^7^ cells id in biweekly intervals followed by bimonthly vaccinations	all pts: local reaction at injection site (grade I) 1 pt pain in draining lymph nodes 1 pt drop of platelet count after 1st vaccination 1 pt flare up of pre-existing inflammation of the Achilles tendon
[67]	12	HCV-related hepato-carcinoma	EP with HSP70 mRNA	3 times 10^7^ to 3 × 10^7^ with 3 week interval	1 pt: grade I: ALT/AST increase 3 pts grade II: hyperglycemia, ALT increase, ALT/AST increase 1 pt grade III liver abscess (not treatment related)
[72]	10	HIV infection	EP with Gag, Vpr, Rev and Nef mRNA	10^7^ id in monthly intervals for 4 cycles	6 patients with either fatigue (grade I), or local reaction at injection site (grade I), flu-like-symptoms (grade I), one pt with each: headache (grade I), diarrhea (grade I), axillary pain (grade I), RF increase (grade I). nausea (grade 1), increase in creatinine (grade I), hematochezia (grade I), eye inflammation (grade I), insomnia (grade I), SCC (grade II), reflux (grade II), GI pain (grade III), appendicitis (grade III), anemia (grade I)
[74]	17	HIV infection	EP with Tat-DC-Lamp, Rev-DC-Lamp or Nef-DC-Lamp mRNA	3 × 10^7^ cells sc and id 4 times in 4 week intervals	16 pts: local reactions (grade I)
[75]	6	HIV infection	Gag-DC-Lamp or Tat-Rev-Nef-DC-Lamp mRNA	10^7^ cells sc (50%) and id (50%) 4 times in monthly intervals	1 pt fever 6 pts local reaction
[68]	7	4 healthy volunteers, 3 HSCT recipients	EP with CMV pp65 mRNA	4 times 10^7^ (HTSC-patients) or 10^5^ (HV) id at weekly intervals	7 pts local reaction grade II (all) 2 HVs headache grade I 1 HV myalgia grade I 1 pt moderate gastrointestinal GVHD grade II (HSCT)
[71]	10	HIV infection	EP with Gag and Nef mRNA	4 × 5 × 10^6^–15 × 10^6^ DCs at week 0,2,6,10	no AEs larger grade II
[73]	35	HIV infection	EP with Gag, Vpr, Rev, and Nef mRNA	4 id-injections of at least 10^7^ DCs with 4 week intervals	25 pts local reactions (grade I) possibly related: headache, nausea, depression dizziness vivid dreams lymphadenopathy, rashes

m: metastatic, pt(s): patient(s), ns: not specified, St.: stage, EP: electroporation, HSCT: hematopoietic stem cell transplantation, aT-RNA: autologous tumor RNA, aTSC-RNA: autologous tumor stem cell RNA, AML: acute myeloid leukemia, MAGE: melanoma-associated antigen, Lamp: lysosome-associated membrane protein, hTERT: human telomerase reverse transcriptase, BCMA: B-cell maturation antigen, MUC1: mucin 1, CEA: carcinoembryonic antigen, PSA: prostate-specific antigen, PAP: prostatic acid phosphatase, WT1: Wilms Tumor 1, CMV: cytomegalovirus, HSP70: heat-shock protein 70, Gag: HIV group-specific antigen, Vpr: HIV viral protein R, Rev: HIV reverse transcriptase, Nef: HIV negative regulatory factor, Tat: HIV trans-activator of transcription, IP: immunoproteasome, ns: not specified, AE: adverse event, GVHD: graft versus host disease, ALT: alanine transaminase, AST: aspartate transaminase, RF: rheumatoid factor, ANA: anti-nuclear Ab, HV: healthy volunteer, ICB: immune checkpoint blockade, id: intradermally, iv: intravenously, sc: subcutaneously, i.n.: intranodally.

**Table 4 pharmaceutics-12-00092-t004:** Active clinical trials with RNA-transfected DCs (from clinicaltrials.gov; status December 2019).

NCT-Number	Country	Title	Antigen	Transfection *	Phase	Status
NCT01983748	Germany	Dendritic Cells Plus Autologous Tumor RNA in Uveal Melanoma	aT-RNA	EP	III	recruiting
NCT03615404	USA	Cytomegalovirus (CMV) RNA-Pulsed Dendritic Cells for Pediatric Patients and Young Adults with WHO Grade IV Glioma, Recurrent Malignant Glioma, or Recurrent Medulloblastoma	CMV-pp65-LAMP	pulsed	I	active, not recruiting
NCT02405338	Norway	DC Vaccination for Post-remission Therapy in AML	WT1 Prame	transfected	I/II	active, not recruiting
NCT02465268	USA	Vaccine Therapy for the Treatment of Newly Diagnosed Glioblastoma Multiforme	CMV pp65-LAMP	pulsed	II	recruiting
NCT02649582	Belgium	Adjuvant Dendritic Cell-Immunotherapy Plus Temozolomide in Glioblastoma Patients	WT1	loaded	I/II	recruiting
NCT01456104	USA	Immune Responses to Autologous Langerhans-Type Dendritic Cells Electroporated with mRNA Encoding a Tumor-Associated Antigen in Patients With Malignancy: A Single-Arm Phase I Trial in Melanoma	mTRP2	EP	I	active, not recruiting
NCT03083054	Brazil	Cellular Immunotherapy for Patients with High Risk Myelodysplastic Syndromes and Acute Myeloid Leukemia	WT1	EP	I/II	active, not recruiting
NCT04157127	USA	Th-1 Dendritic Cell Immunotherapy Plus Standard Chemotherapy for Pancreatic Adenocarcinoma	ns	loaded	I	not yet recruiting
NCT01995708	USA	CT7, MAGE-A3, and WT1 mRNA-electroporated Autologous Langerhans-type Dendritic Cells as Consolidation for Multiple Myeloma Patients Undergoing Autologous Stem Cell Transplantation	Mage-A3, Mage-C1, WT1	EP	I	active, not recruiting
NCT01197625	Norway	Vaccine Therapy in Curative Resected Prostate Cancer Patients	aT-RNA, hTERT, survivin	loaded	I/II	active, not recruiting
NCT01686334	Belgium	Efficacy Study of Dendritic Cell Vaccination in Patients with Acute Myeloid Leukemia in Remission	WT1	EP	II	recruiting
NCT03548571	Norway	Dendritic Cell Immunotherapy Against Cancer Stem Cells in Glioblastoma Patients Receiving Standard Therapy	aTSC-RNA, survivin, hTERT	transfected	II/III	recruiting
NCT02366728	USA	DC Migration Study for Newly Diagnosed GBM	CMV pp65-LAMP	pulsed	II	active, not recruiting
NCT02808416	China	Personalized Cellular Vaccine for Brain Metastases (PERCELLVAC3)	ns	pulsed	I	active, not recruiting
NCT02649829	Belgium	Autologous Dendritic Cell Vaccination in Mesothelioma	WT1	loaded	I/II	recruiting
NCT00639639	USA	Vaccine Therapy in Treating Patients with Newly Diagnosed Glioblastoma Multiforme	CMV pp65-LAMP	loaded	I	active, not recruiting
NCT02709616	China	Personalized Cellular Vaccine for Glioblastoma (PERCELLVAC)	individually selected TAAs	pulsed	I	active, not recruiting
NCT03927222	USA	Immunotherapy Targeted Against Cytomegalovirus in Patients with Newly Diagnosed WHO Grade IV Unmethylated Glioma	CMV pp65-LAMP	pulsed	II	recruiting

* unfortunately, some researchers do not specify the method of transfection, although most probably electroporation was used. GBM: glioblastoma multiforme, aT-RNA: autologous tumor RNA, aTSC-RNA: autologous tumor stem cell RNA, AML: acute myeloid leukemia, MAGE: melanoma-associated antigen, CT7: MAGE-C1, Lamp: lysosome-associated membrane protein, hTERT: human telomerase reverse transcriptase, WT1: Wilms Tumor 1, CMV: cytomegalovirus, LAMP: lysosome-associated membrane protein, mTRP2: murine tyrosinase-related peptide 2, ns: not specified, EP: electroporation.

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
