# Peer review of "Therapeutic Cancer Vaccination with Ex Vivo RNA-Transfected Dendritic Cells—An Update"

_pharmaceutics, 2020, doi:10.3390/pharmaceutics12020092_

Round 1
Reviewer 1 Report
Authors elegantly described a comprehensive review of therapeutic cancer vaccination with ex-vivo RNA-transfected dendritic cells (DCs).
1. It is understandable to mention methodological details with RNA-transfection to DCs in sessions of 3.3 and 3.4; as various approach using electroporation, pulse, load and transfection are described in Table 2.
2. Authors introduced new approach using LAMP-1 and DC-LAMP to overcome the limited presentation in HLA class II, on the other hand, another technique independent on exogenous cytokines for DC manufacturing is explained. It is better to explain optimized RNA-transfection of DCs as shown in a Figure.
3. Cancer immunotherapy is making forward to individualized treatment based on the gnomic medicine targeting neoantigens. Neoatigen vaccination is expected for future cancer immunotherapy, therefore, it is better to provide advancement of neoantigen RNA-transfected DC vaccination.
4. In conclusion on the basis of review of the literature, it is better to conclude authors' opinions for the prospect of ex-vivo RNA-transfected DCs for clinical trials using optimized protocol to evaluate clinical efficacy in combination with the therapies including immune checkpoint inhibitors.
5. References should be sited correctly as ref 76 and 77 are displayed reversely in Table 1.
Reviewer 2 Report
Dendritic cells have been used as cancer vaccine in several clinical trials, alone or in combination with drugs, and in this review, there is a detailed description of them, enumerating the antigens, the diseases treated, the route of administration and the doses, with complete report of side effects and clinical response. It is a very interesting topic, because, as the authors state, “it is very optimistic to expect DC vaccines by themselves to frequently produce significant clinical benefit in the setting of established late stage malignancies like stage IV melanoma” but, their use in combination with other treatments can open new perspectives of therapies.
In my opinion, to complete the scenario, will be useful to the readers enumerating also the drawbacks of this vaccine, such as the cost to be sustained to produce a personalized therapy, and the high specialization of the staff involved in the manipulation and reinfusion of the DC in patient, that can be recruited only by big health institutions, reducing the possibility to administrate these therapies widely.
Reviewer 3 Report
In this manuscript, authors have extensively reviewed the usage of ex vivo RNA-transfected dendritic cells for therapeutic cancer vaccination. This article provides good knowledge on the current clinical trials and adverse events, which could be a great addition to the current literature.
Authors should address the following concerns:
An addition of a schematic showing dendritic cell- based cancer vaccination approaches with an emphasis on ex vivo RNA-transfection would help the readers to understand the concept. Table 3 and Table 4 need to be reorganized. For example: Each disease type could be organized together and so on. Also, all the acronyms should be included below the table. Future directions including alternative approaches, requirement of combinatorial treatments, state of the art design methodologies, choice of route of administration and management of adverse events should be discussed elaborately with any relevant references.Author Response
Please see the attachment

Round 2
Reviewer 1 Report
Authors elegantly revised a comprehensive review of therapeutic cancer vaccination with ex-vivo RNA-transfected dendritic cells.
Readers would be interested in this review article for their research fields.
Reviewer 2 Report
The new paragraph added “4.3. Challenges and future perspectives of DC vaccine therapy" fulfilled the request to discuss the drawback and the future of this therapy
Reviewer 3 Report
Authors have adequately addressed the comments and made necessary edits.